# Emerging Roles of RNA 3′-end Cleavage and Polyadenylation in Pathogenesis, Diagnosis and Therapy of Human Disorders

**DOI:** 10.3390/biom10060915

**Published:** 2020-06-17

**Authors:** Jamie Nourse, Stefano Spada, Sven Danckwardt

**Affiliations:** 1Institute for Clinical Chemistry and Laboratory Medicine, University Medical Center of the Johannes Gutenberg University, 55131 Mainz, Germany; James.Nourse@unimedizin-mainz.de (J.N.); Stefano.Spada@unimedizin-mainz.de (S.S.); 2Center for Thrombosis and Hemostasis (CTH), University Medical Center of the Johannes Gutenberg University, 55131 Mainz, Germany; 3German Center for Cardiovascular Research (DZHK), Rhine-Main, Germany

**Keywords:** 3′ end processing, cleavage and polyadenylation, alternative polyadenylation, cardiovascular disorders, neurodegenerative disorders, cancer, disease, biomarker, therapy

## Abstract

A crucial feature of gene expression involves RNA processing to produce 3′ ends through a process termed 3′ end cleavage and polyadenylation (CPA). This ensures the nascent RNA molecule can exit the nucleus and be translated to ultimately give rise to a protein which can execute a function. Further, alternative polyadenylation (APA) can produce distinct transcript isoforms, profoundly expanding the complexity of the transcriptome. CPA is carried out by multi-component protein complexes interacting with multiple RNA motifs and is tightly coupled to transcription, other steps of RNA processing, and even epigenetic modifications. CPA and APA contribute to the maintenance of a multitude of diverse physiological processes. It is therefore not surprising that disruptions of CPA and APA can lead to devastating disorders. Here, we review potential CPA and APA mechanisms involving both loss and gain of function that can have tremendous impacts on health and disease. Ultimately we highlight the emerging diagnostic and therapeutic potential CPA and APA offer.

## 1. Introduction

In multicellular organisms almost every cell contains the same genome, yet complex spatial and temporal diversity is observed in gene transcripts. This is achieved through multiple levels of processing leading from gene to protein, of which RNA processing is an essential stage [1]. Following transcription of a gene by RNA polymerases to produce a primary RNA transcript, further processing is required to produce a stable and functional mature RNA product. This involves various processing steps including RNA cleavage at specific sites, intron removal, or splicing to substantially increase the transcript repertoire [2]. Moreover, a crucial feature of the RNA processing of most genes is the generation of 3′ ends through an initial endonucleolytic cleavage, followed in most cases by the addition of a poly(A) tail, a process termed 3′ end cleavage and polyadenylation (CPA, see Figure 1) [3]. The poly(A) tail ensures the translocation of the nascent RNA molecule from the nucleus to the cytoplasm [4], enhances translation efficiency [5] and controls RNA degradation [6], and thereby ultimately governs the production of a protein [7]. Thus, a full appreciation of CPA as a regulatory entity is crucial for understanding numerous aspects of gene expression. As the mechanistic features of 3′ end formation have been thoroughly reviewed [3,8,9,10,11,12] only the key features will be described here.

### 1.1. Cleavage and Polyadenylation Factors Interacting with Poorly Conserved Sequence Motifs Control mRNA 3′ end Formation

CPA is carried out by a multi-subunit 3′ end processing complex, which involves over 80 trans-acting proteins [14], comprised of four core protein subcomplexes (Figure 1A). These consist of (1) cleavage and polyadenylation specificity factor (CPSF), comprised of proteins CPSF1-4, factor interacting with PAPOLA and CPSF1 (FIP1L1), and WD repeat domain 33 (WDR33); (2) cleavage stimulation factor (CstF), a trimer of CSTF1-3; (3) cleavage factor I (CFI), a tetramer of two small nudix hydrolase 21 (NUDT21) subunits, and two large subunits of CPSF7 and/or CPSF6; and (4) cleavage factor II (CFII), composed of cleavage factor polyribonucleotide kinase subunit 1 (CLP1) and PCF11 cleavage and polyadenylation factor subunit (PCF11). Additional factors include symplekin, the poly(A) polymerase (PAP), and the nuclear poly(A) binding proteins such as poly(A) binding protein nuclear 1 (PABPN1) [3,10].

CPA is initiated by this complex recognising specific cis-element sequences within the nascent pre-mRNA transcripts termed polyadenylation signals (PAS) [3]. The PAS sequence normally consists of either a canonical AATAAA hexamer, or a close variant usually differing by a single nucleotide (e.g., ATTAAA, TATAAA). It is located 10 to 35 nucleotides upstream of the cleavage site (CS) usually consisting a CA dinucleotide [15,16]. The PAS is also determined by surrounding auxiliary elements, such as upstream U-rich elements (USE), or downstream U-rich and GU-rich elements and G-rich sequences (DSE) [17].

As soon as the nascent RNA molecule emerges from RNA polymerase II (RNA Pol II), the CPSF complex is recruited [18] to the PAS hexamer [14,19,20] through CPSF4 and WDR33 [21,22,23,24], and the USE through FIP1L1 [20,25]. The CstF complex recognises the DSE [26,27,28] via CSTF2 directly interacting with the RNA [29,30,31]. The CFI complex is recruited to the USE with NUDT21 binding UGUA elements and CPSF6 and CPSF7 also contacting RNA [32,33]. Finally, CFII binds downstream G-rich sequence elements via PCF11, with CLP1 being required for the cleavage activity of CFII [34]. Upon successful assembly of this macromolecular machinery, CPSF3 performs the endonucleolytic cleavage [35,36] followed by a non-templated addition of approximately 50-100 A residues [37].

The proper formation of the poly A tail in conjunction with the 3′ untranslated regions (3′UTR) in the transcript directs stability, nuclear export, sub-cellular localisation and translation efficiency via cis-acting elements interacting with trans-acting factors, such as microRNAs (miRNA) or RNA-binding proteins (RBP) in given cellular contexts [38,39,40,41]. In addition, 3′UTRs direct protein localisation through scaffolds where protein–protein interactions are established by one of the interaction partners being recruited by a 3′UTR [42,43].

The intricate, bidirectional coupling of 3′ end formation to transcription and other co-transcriptional events, such as capping and splicing [12], ensures that polyadenylation occurs timely and at the correct position. At the same time, this intricate net of interactions provides numerous mechanisms for gene regulation. The efficiency of CPA is modulated by coupling with components of the transcription [12,44,45], splicing [46,47,48,49], epigenetic [50,51,52,53,54], and signalling [55] machineries. Additionally, CPA can provide mechanisms for gene regulation through variations in the efficiency of CPA processing resulting from changing factor levels in trans [56,57]. Furthermore, this is determined by cis PAS sequence elements of individual genes [58,59].

### 1.2. Alternative 3′ end Formation Expands Transcriptional Complexity

Alternative polyadenylation (APA) occurs when more than one PAS is present within a pre-mRNA and provides an additional level of complexity in CPA-mediated RNA processing (Figure 1B). Early studies revealed a significant portion of genes undergo APA [60,61], and with the advent of next-generation RNA sequencing technologies [62,63] the large scale regulation of genes has become apparent, with approximately 70% of the transcriptome exhibiting APA regulation [64,65]. As APA determines 3′UTR content and thus the regulatory features available to the mRNA, changes in the APA profile of a gene can have enormous impacts on expression (further detailed below).

Approximately 80% of all APA events occur in the 3′UTR [13], resulting in transcripts that differ only in 3′UTR length. However, in the remaining 20% an alternative PAS is used upstream of the last terminal exon, often in an alternatively spliced intron. Use of such a PAS produces transcript isoforms with alternative coding regions (a process termed internal APA) regulating the functional properties of the resulting protein [3,60] (Figure 1B). APA thereby profoundly expands the functional diversity by affecting almost all genes.

### 1.3. Cleavage and Polyadenylation Signal Motifs Determine Alternative Polyadenylation

The intrinsic strength of a PAS, as regulated by its affinity to CPA factors, determines PAS choice and subsequently APA. This can be modulated by additional elements, for example G-rich DSEs can enhance CPA at a particular PAS [66,67]. In addition there is an interplay with transcription termination and RNA Pol II velocity, with the PAS not only directing 3′ end processing but dictating changes in the rate and extent of transcription, with RNA Pol II slowing down or pausing upon traversing a PAS [68]. This pausing is decisive in PAS choice [69], where if an apparently strong distal site is not yet transcribed it cannot be recognised by the CPA machinery. Accordingly, reduction in RNA Pol II elongation rates has been observed to result in preferential use of a proximal PAS [70]. Motifs tend to be enriched at distal sites, where canonical hexamers are also more common [16]. This allows read-through of weaker proximal sites. These tend to be suboptimal and to contain variant hexamers [15,16,60], and are used only when the concentrations of 3′ end processing factors are high.

### 1.4. Cleavage and Polyadenylation Factor Abundance Regulates Alternative Polyadenylation

As described previously, CPA and APA are tightly coupled to transcriptional processes involved in the production of a mature RNA, such as capping and splicing [12,71]. Additionally, CPA and subsequently APA are coupled to other regulatory events such as signalling pathways controlling RNA-binding of the CPA machinery [55], RNA polymerase II kinetics and termination efficiency [70,72], and epigenetic control, including DNA methylation [54], chromatin structure [53,73,74], and RNA methylation [51,75,76]. These interactions play significant roles in PAS selection resulting in APA.

Additionally, the abundance of individual 3′ end processing factors can affect APA at the global level (Figure 1C). CFI has a particularly strong impact on PAS choice through preferential binding to the distal PAS, enhancing its use [13,25,77,78,79,80], with reduction of the CFI component NUDT21 being linked to 3′UTR shortening [78]. The CPSF complex contributes to PAS selection [25,81] through CSTF2 targeting proximal non-canonical PASs possessing both upstream adenosine-rich regions and an atypical CSTF2-binding motif downstream of the PAS [82]. Depletion of CSTF2 results in increased use of the distal PAS, particularly when the CSTF2 tau variant (CSTF2T) is co-depleted [80,81]. CFII also promotes proximal PAS usage [79] and depletion of PCF11 results in enhancement of distal PAS usage [13], consistent with the role of PCF11 in modulating RNA Pol II processivity and transcription termination [83].

Finally, transcripts exhibiting dynamic regulation at the 3′ end are typically encoded by phylogenetically ancient genes, which corresponds to the phylogenetic age of most executing APA regulators [13]. Such phylogenetically conserved genes appear to regulate basic processes that when dysregulated result in more severe phenotypes [84]. Together, these interactions illustrate the central role of CPA in the complex crosstalk between various cellular processes in the control of transcriptome diversity.

## 2. The Role of Cleavage and Polyadenylation in Disease

Recent studies in human pathologies have shown genetic variants affecting RNA processing are as frequent, and largely independent from, variants affecting transcription [1]. In disease, CPA can be dysregulated through alterations in cis affecting sequence elements transcribed into the nascent pre-mRNA, or in trans by affecting the regulatory proteins executing CPA (Figure 2, Table 1). Additionally, in response to dynamic and pathological environmental changes, CPA regulatory proteins may modulate PAS choice and thereby direct APA [13,85]. In the following sections we will discuss some prototypical findings of a larger compilation of disorders (Table 1) resulting from CPA perturbations.

### 2.1. Defects in Cis Resulting in Altered Cleavage and Polyadenylation

Variations from the canonical PAS hexamer (AATAAA) generally reduce CPA efficiency [58,60], although PASs can exhibit sequence flexibility and additional auxiliary sequences may compensate for the loss of core sequence integrity [228]. In the context of genes possessing a single PAS, mutations in this site will alter gene expression, while in genes with multiple, functionally competing, PASs, alterations in APA can result.

#### 2.1.1. Loss of Function Alterations in the Hexamer of a Single PAS Gene Can Affect Gene Expression and Result in Disease

Mutations within the sole PAS of genes were the first alterations in CPA demonstrated to result in disease. In thalassemia patients, initial studies identified a non-deletion mutation with homozygotes suffering from moderate to severe α-thalassemia [229] and resulting in elongated RNA transcripts [182]. This was subsequently found to result from a single nucleotide change in the canonical PAS of hemoglobin subunit alpha 2 (HBA2) [230] with further mutations in this PAS being subsequently observed, all resulting in moderate to severe disease [231,232,233]. In a similar manner, mutations in the canonical PAS of HBB hemoglobin subunit beta (HBB) were also found to result in very low expression levels and to produce extended transcripts [183]. As with HBA2, further HBB PAS mutations were identified, resulting in mild β-thalassaemia [184,234,235,236,237,238,239,240,241,242].

Both HBA2 and HBB utilise a single canonical PAS and as such mutations producing the much less efficiently processed AATAAG [182,184] and AACAAA [183] variant hexamers [58] result in many nascent transcript molecules failing to be cleaved at this site, allowing potential cryptic downstream PASs to be used. Splicing of these extended transcripts occurring in the 3′UTR can result in nonsense-mediated mRNA decay of the transcript, reducing RNA expression. Additionally, a PAS mutation can not only reduce expression of a gene, but read-through interference can affect a downstream gene. For example, HBA2 read through into the downstream HBA1 hemoglobin subunit alpha 1 (HBA1) gene inhibits HBA1 gene expression leading to haemoglobin-H disease, despite the HBA1 gene being intact [243,244].

Other PAS alterations affecting a sole canonical hexamer have since been identified (Figure 2, Table 1). For example, the rs78378222 polymorphism in the tumour suppressor tumor protein p53 (TP53) is associated with an increased risk of overall cancer [112,146,147]. The polymorphism results in the rare variant AATACA hexamer and impairs proper transcription termination and polyadenylation, leading to reduced expression [147]. In the autoimmune polyendocrine IPEX syndrome, alteration of forkhead box P3 (FOXP3) to the rare variant AATGAA results in low mRNA levels and absence of CD4^+^CD25^+^ T regulatory cells [186]. Where alteration of a sole canonical PAS produces a less common variant, lower levels of expression are generally observed. In neonatal diabetes, alteration of the insulin gene PAS to the extremely inefficiently processed hexamer AATAAG results in very low mRNA levels [179]. This has also been documented for the IL2RG PAS altered in severe combined immunodeficiency [189].

Variant hexamers are found in 30% of genes possessing a single PAS [60]. Alterations in these hexamers produce more variable outcomes than alterations in canonical hexamers. The variant ATTAAA hexamer is found in 15% of genes [60] and directs relatively efficient CPA (~70%) as compared to the canonical hexamer [58]. Mutation of this sole PAS in the α-galactosidase A (GLA) gene has been reported in the lysosomal storage disorder Fabry disease and results in transcripts with differing 3′ lengths. Although these transcripts appear to be stable, differences in protein localisation were observed [217]. In contrast, in children with bone fragility disorder, mutations in the single variant PAS of bone morphogenetic protein 1 (BMP1), found in only 3.7% of genes and resulting in a CPA efficiency of approximately 30% of the canonical hexamer [58,60], resulted in low but detectable BMP1 mRNA levels and total absence of protein [177]. This suggests CPA alterations not only affect mRNA stability but they may also mediate an additional effect on translation.

#### 2.1.2. Loss of Function Alterations in a Hexamer of a Multi PAS Gene Resulting in Alternative Polyadenylation and Disease

The above section described alterations in the hexamer of genes possessing a single PAS resulting in impaired CPA and reduced gene expression. However for the vast majority of genes that harbour multiple PAS alterations, causing reduced efficiency of one PAS can result in increased use of other competing PASs, ultimately resulting in APA. This can have varying outcomes depending on the relative positions of the affected PAS and other elements within the alternative 3′UTRs produced (Figure 1B). This is particularly evident in genes in which the proximal PAS contains a canonical hexamer. For example, in patients suffering syndromic microphthalmia, various mutations in the proximal canonical PAS of the N-alpha-acetyltransferase 10, NatA catalytic subunit (NAA10) gene result in isoform lengthening from increased use of a distal rare AAATAA PAS, leading to decreases in mRNA expression of approximately 50% [175]. Similarly, in systemic lupus erythematosus (SLE), a SNP (rs6598) in the proximal canonical PAS of GTPase, IMAP family member 5 (GIMAP5) produces the rare AATAGA hexamer. This results in increased levels of longer transcripts, lower mRNA expression levels and an increased likelihood of developing thrombocytopenia [193]. Alterations in a rare proximal PAS have also been shown to result in transcript lengthening. In leukodystrophy pseudo-deficiency, a mutation in the predominate yet rare AATAAC proximal PAS of arylsulfatase A (ARSA) to AGTAAC results in increased use of distal canonical PASs and reduced expression, potentially through the lengthened 3′ UTR allowing miRNA- or RBP-mediated silencing [221].

Interestingly, as opposed to the transcript lengthening observed when polymorphisms or mutations in proximal PAS hexamers reduce CPA efficiency, alterations in distal PAS hexamers predominately result in 3′UTR shortening. An example of this is seen with the serotonin transporter gene, solute carrier family 6 member 4 (SLC6A4), which possesses two 3′UTR PASs [245,246], with the proximal being a variant AATGAA hexamer with low CPA efficiency [58], while the distal contains a common SNP (rs3813034) resulting in either of the very rare hexamers ATTAAC or AGTAAC. The G-allele leads to less efficient use of the distal PAS. This correlates with total SLC6A4 expression and has been associated with an increased risk for anxiety disorders [215,216] and suicidal behaviour [225]. This suggests the distal sequence stabilises SLC6A4 [215], possibly by heterogeneous nuclear ribonucleoprotein K (hnRNPK) binding of the distal PAS and reducing miRNA-16 binding [214]. In Zellweger spectrum disorder, increased expression of peroxisomal biogenesis factor 6 (PEX6) results from a deletion in the 3′UTR of PEX6, which eliminates the distal canonical PAS leaving only the shorter isoform produced from the remaining proximal PAS. In combination with other mutations resulting in a pathogenic protein, the increased level of mutated PEX6 protein resulting from 3′UTR shortening impaired the function of the PEX1–PEX6 complex, resulting in defective import of peroxisomal proteins [227].

#### 2.1.3. Gain of Function Alterations Can also Affect Cleavage and Polyadenylation and Result in Disease

Albeit far less commonly reported, mutations resulting in the de novo creation of cryptic PAS have been identified (Figure 2, Table 1). In several types of cancer, including mantle cell lymphoma, a translocation often leads to over-expression of Cyclin D1 (CCND1) [247]. However, patients have been identified where point mutations create a novel canonical PAS leading to CCND1 3′UTR shortening, resulting in transcript and protein over-expression due to removal of an AU-rich element and miR-16-1 binding sites [133]. Additionally, PAS gain of function mutations have been described in two disorders associated with expansion of repeats. Huntington disease is caused by a CAG repeat expansion in exon 1 of the huntingtin (HTT) gene [248]. This expanded CAG repeat facilitates binding of serine and arginine rich splicing factor 6 (SRSF6) to the transcript resulting in the promotion of polyadenylation from a cryptic PAS within intron 1 and production of a highly pathogenic truncated transcript [220]. In facioscapulohumeral muscular dystrophy (FSHD), shortening of a 4q35 sub-telomeric D4Z4 repeat array to less than 10 copies is associated with FSHD and results in reduced methylation, subsequent chromatin remodelling, and increased transcription of an ORF within the repeat encoding the double homeobox 4 (DUX4) protein [249,250]. These transcripts appear to be unstable due to the absence of a PAS in internal D4Z4 units, however SNPs distal to the last D4Z4 repeat create a canonical PAS for the distal D4Z4 repeat [197]. This stabilises the transcript, allowing expression of the DUX4 protein, which has been associated with FSHD [251]. Alterations can also create a gain of function, increasing PAS processing efficiency. When this affects one PAS within a multi-PAS gene, APA can be altered. This can be seen with the interferon regulatory factor 5 (IRF5) SNP rs10954213, where the A allele is a risk factor for SLE and produces a canonical PAS, while the G allele produces a variant AATGAA PAS [191]. Presence of a canonical hexamer at the proximal PAS shifts polyadenylation away from the distal canonical PAS, leading to increased expression, which appears to result from the loss of AU-rich elements from the long isoform [190,192].

Finally, gain of function variations can arise from altered RNA editing of transcripts. In the neurological disorder amyotrophic lateral sclerosis (ALS), increased RNA editing within an intron of the solute carrier family 1 member 2 (SLC1A2) glutamate transporter creates a cryptic PAS resulting in intron retention with termination of transcription transcripts [212]. Reduced levels of SLC1A2 glutamate transporter have been observed in ALS [252,253]. Considering the previously underestimated wide repertoire of RNA editing events in humans [254,255], this finding has important implications in that mere genome sequencing will not identify such disease-eliciting perturbations (see also diagnostic section below).

#### 2.1.4. Alterations in Non-Hexamer Elements Affecting Cleavage and Polyadenylation and Resulting in Disease

Although other CPA sequences are considered to be much more tolerant to alterations than the PAS hexamer [256], alterations in these sequences can also result in disease. This was first demonstrated by the prominent coagulation factor II, thrombin (F2) 20210A gain of function mutation altering the cleavage site of the prothrombin gene from the less efficient CG to the predominant and mechanistically most efficient CA dinucleotide [171,174,257]. This significantly increases cleavage efficiency, leading to enhanced mRNA and protein expression, and has been linked to thrombophilia [258]. Other mutations also affect CPA of the prothrombin gene. The rare gain of function C20221T [259] mutation, 11 nucleotides downstream of the cleavage site in the putative CSTF binding site, promotes the efficiency of CPA [174] and has been found in patients with abnormal thrombosis [174]. The C20209T [260] mutation located at the penultimate position of the 3′UTR stimulates CPA and up-regulates prothrombin protein expression [173]. However, these effects are presumably gene-specific; the prothrombin gene utilises an unusual architecture where a USE compensates for the weak activity of the cleavage site and downstream U-rich element, which displays an unusually low density of uridine residues when compared to efficiently 3′ end processed mRNAs. Here the USE controls PAS usage in response to stress, demonstrating the mechanistic links between signalling in response to environmental cues and modulation of CPA in the nucleus [261].

Alterations in DSE have been associated with disease. The gamma chain of human fibrinogen (FGG) exists in 2 isoforms, FGG-γA formed by all 10 exons, and FGG-γ’ formed by the use of a PAS in intron 9 resulting in a truncated protein consisting of exons 1–9 and the first 60 nucleotides of intron 9 [262,263]. A relatively common C to T polymorphism in the DSE has an impact on the putative CstF64 binding site, altering APA, the subsequent fibrinogen γ’/γA isoform ratio and increasing the risk for deep venous thrombosis [169,170]. The Na^+^ and K^+^ cotransporter ATPase Na+/K+ transporting subunit beta 1 (ATP1B1) undergoes APA due to the use of multiple PASs with the 3′UTR [264,265], with short 3′UTR isoforms being translationally more efficient due to a translational repressor sequence in the region unique to the long isoform [266,267,268]. A common human polymorphism (rs12079745) has been identified in the DSE of the proximal PAS and is strongly associated with high blood pressure [269]. Variations in this polymorphism mediate changes in the relative abundance of ATP1B1 3′ isoforms by regulating APA of the gene [167].

#### 2.1.5. Alterations outside Characterised Elements Affecting Cleavage and Polyadenylation and Resulting in Disease

While alterations in established CPA elements have clear functional outcomes, changes in less defined regions of 3′UTRs affecting CPA have been reported. A SNP (ss52051869) in the 3′UTR of the arginine transporter solute carrier family 7 member 1 (SLC7A1) is involved in predisposition to essential hypertension and has been associated with altered protein expression [166]. SLC7A1 is alternatively polyadenylated at two sites, resulting in APA isoforms with varying 3′UTR lengths. The relative expression of these isoforms correlates with the allele frequency of the SNP, which alters a binding site for the transcription factor Sp1 [165]. The long 3′UTR isoform of SLC7A1 exhibits a lower level of expression, regardless of allele status, and this appears to result from the presence of microRNA 122 (miR-122) binding sites at the long 3′UTR of SLC7A1, affecting translation and stability [270,271].

A similar occurrence is found in Parkinson’s disease (PD). The brains of PD patients contain cytoplasmic protein aggregates largely composed of α-synuclein (SNCA) [272]. SNCA mRNAs possess four functional PASs [273] and in PD an increased use of the distal PAS results in longer isoforms, which preferentially localise to mitochondria and are more likely to aggregate [224]. Disease-associated SNP variants within the long SNCA 3′UTRs enhance protein accumulation and mitochondria localisation [224]. An additional SNP, located between the two most frequently used PASs, is located immediately upstream of, but not within, a DSE, which could affect CPA efficiency at the immediately upstream PAS [273].

In addition, larger 3′UTR lesions have also been reported to affect CPA. This is exemplified in Lynch syndrome, which confers increased risks of multiple cancers and results from mutations in DNA mismatch repair genes, including mutS homolog 6 (MSH6) [274]. A 20 bp sequence found to be duplicated in the vicinity of the MSH6 sole canonical PAS in Lynch syndrome patients results in reduced CPA and expression [132].

Approximately 20% of all APA events fall in the coding region, thereby resulting in proteins with distinct regulatory or functional properties [13] (Figure 1B). Along these lines, alterations within introns have also been reported to affect CPA. The RET proto-oncogene possesses multiple PASs, which result in multiple 3′ isoforms [275], and expression of these isoforms has been shown to be altered in endocrine tumours [137]. An internal PAS is positioned within an intron, six nucleotides downstream of a C/T polymorphism, which is within a potential binding site for the PBX homeobox 1 (PBX1) transcriptional factor. C/T heterozygotes were found to comprise a significantly higher percentage of endocrine tumour patients, while T/T homozygotes were found exclusively in endocrine tumours with high malignant potential [137]. As the T allele represents the canonical binding motif for PBX1 [276], these results suggest PBX1 binding may enhance CPA at the adjacent PAS, leading to higher expression levels of RET.

Finally, seeming unrelated alterations can have effects on CPA. The fragile X mental retardation 1 gene (FMR1) produces multiple APA transcript isoforms in Fragile X syndrome (FXS) and related diseases, such as fragile X-associated immature ovarian insufficiency [277,278] and fragile X-associated tremor and/or ataxia syndrome [279,280]. FXS results from a CGGn expansion in the 5′UTR of FMR1, with greater than 200 repeats resulting in methylation-coupled transcriptional silencing [281,282,283]. When FMR1 presents permutation alleles in the 5′UTR, consisting of CGG repeats that can be extended from 55 to 200 and do not inactivate the gene [277,278], FMR1 mRNA isoforms derived from two variant PASs decrease, contributing to a reduction of FMR1 expression levels [218].

The presence of 3′UTR alterations that functionally affect CPA/APA outside established CPA elements indicates a lack of knowledge of how 3′UTR sequences are involved in regulating CPA and APA. As 3.7% of the genetic variants detected in GWAS studies are localised within UTRs [284,285] and whole-genome sequencing has identified substantial UTR functional deregulation occurring in disease [286,287,288], it is imperative to unravel these mechanisms to fully understand the impact of CPA and APA on disease.

### 2.2. Defects in Trans Resulting in Altered Cleavage and Polyadenylation

As described in the preceding sections, many alterations in cis, both at the DNA and RNA level, can result in aberrant CPA and ultimately to disease by affecting individual genes. However, as can be seen in the following sections, global alterations affecting CPA and APA in trans are a significant feature of many diseases.

#### 2.2.1. Alterations in Trans Factors in Alternative Polyadenylation and Disease

APA is modulated in various normal physiological circumstances [3,13,289]. APA regulation plays important roles in the immune system [154], neural systems [290], stem cell differentiation [291], and development [292]. Specific tissues appear to follow global patterns either favouring shorter or longer 3′UTR isoforms [292,293,294], suggesting APA to play a central role in the establishment of regulated expression networks fundamental to a multitude of biological roles [295].

APA shortening (where a relatively higher amount of transcripts undergo polyadenylation at a proximal PAS) in proliferating cells has been shown to be accompanied by an increased expression of polyadenylation factors [13,141,154,292,296,297], while it has been reported that cancer cell lines are significantly enriched in mRNA containing shortened 3′UTRs relative to non-transformed cells [13,144,154,296,298]. Widespread 3′UTR shortening is consistently reported in cancers [141,144] including breast [91,96,97,100,101], colorectal [108,110], gastric [115], neuroendocrine [157], neuroblastoma [13], and glioblastoma [119]. Shortening has been associated with poor outcome in breast, lung [102,131], pancreatic ductal adenocarcinoma [153], and neuroblastoma [13]. Shortening has also been reported for non-coding RNAs, such as competing endogenous long non-coding RNAs (ceRNA) [94] and small nuclear RNAs (snRNA) [31], and in other conditions including ischemia/reperfusion injury [168] and Alzheimer’s disease [202,203].

This has led to a model where APA shortening of 3′UTR of oncogenes in cancer cells allows evasion of the repressive effects of microRNA and RNA binding proteins [98,115,121,289,298,299]. An estimated 70% of genes have conserved miRNA target sites [64,99] and 11% have mRNA destabilising AU-rich elements (ARE) within their 3′UTRs [300]. Additionally, deregulation of miRNA/ARE-RBPs is associated with many human cancers [301]. When accompanied with increased expression of the oncogene, as a result of its 3′UTR shortening, this supports the highly proliferative phenotype of tumours [78,144]. Other proposed mechanisms involve 3′UTR shortening repressing tumour suppressor genes in trans by disrupting ceRNA crosstalk [94] or releasing miRNAs that would have otherwise bound to longer 3′UTR isoforms [302].

#### 2.2.2. Alterations of the CFI Complex in Alternative Polyadenylation and Disease

The CFI complex is well characterised in the regulation of APA. It has a particularly strong impact on PAS choice through preferential binding to the distal PAS, enhancing its use. For distal non-canonical PASs, the CFI complex can facilitate PAS usage through NUDT21 binding to USE UGUA motifs and stabilise the binding of CPSF complex to the pre-mRNA [25,32,303,304]. Depletion of CFI components NUDT21 [13,25,77,78,79,80,305] or CPSF6 [13,25,82] allows the interaction of CPSF with proximal PASs, resulting in the global shortening of 3′UTRs (Figure 1C). Reduced levels of the CFI component NUDT21 have been reported to result in global 3′UTR shortening in a variety of disorders, particularly in cancers.

In glioblastoma patients, NUDT21 expression is reduced, with lower levels being associated with shortened 3′UTRs and worse survival [78,117]. Supporting a role in 3′UTR shortening, depletion of NUDT21 in glioblastoma cells results in proximal PAS usage as well as increased cell proliferation and tumorigenicity [78]. NUDT21 levels are also reduced in hepatocellular carcinoma [121,122] and bladder cancer [88], and low NUDT21 associates with shorter 3′UTRs and adverse outcome [88,122]. Alterations in signalling pathways are associated with NUDT21-mediated global 3′UTR shortening. In glioblastoma, this involves the Ras pathway [117], while in bladder cancer alterations in the Wnt/β-catenin and nuclear factor kappa B (NF-κB) pathways were found [88]. NUDT21 down-regulation is also associated with other diseases. In idiopathic pulmonary fibrosis patients, NUDT21 is down-regulated, and NUDT21 knockdown in lung fibroblast cells results in a significant shift to proximal PAS usage with enrichment of transforming growth factor-beta (TGF-β), Wnt, and hypoxia inducible factor 1 subunit alpha (HIF1A) signalling pathways [195]. In osteosarcoma cells it has been reported that miR-181a down-regulates the expression of NUDT21, inhibiting proliferation and promoting apoptosis [306].

However, elevated NUDT21 expression is also associated with disease. In individuals with neuropsychiatric disease, copy-number variations producing elevated NUDT21 levels have been reported to result in the increased usage of the distal PAS and reduced expression of ethyl-CpG binding protein 2 (MECP2), a gene closely associated with neuropsychiatric disease [223]. In chronic myelocytic leukemia patients NUDT21 is highly expressed, and depletion of NUDT21 in leukemia cells inhibits growth and proliferation, possibly through inhibition of the extracellular signal-regulated kinase (ERK) signalling pathway [307]. However in this particular study APA changes were not assessed.

Altogether this demonstrates an important role of NUDT21 in various disease entities. This reflects its pervasive role in APA regulation [13] (Figure 1C).

#### 2.2.3. Alterations of the CSTF Complex in Alternative Polyadenylation and Disease

Downstream elements (DSE) assist in the selection of non-canonical PASs [32,228]. CstF interaction with the DSEs of proximal, non-canonical PASs promotes their use [26,27,28]. This appears to be through CSTF2 interacting with atypical binding motifs downstream of the PAS [82]. While depletion of CSTF2 has a relatively small effect on APA, co-depletion of CSTF2 and CSTF2T leads to significant APA shifts, primarily to the distal PAS, which is thought to reflect the general higher efficiency of distal PASs [81]. Conversely, elevated CSTF2 has been reported to increase the use of a weaker proximal PAS [308]. For example the internal APA switch from membrane-bound to secreted form of immunoglobulin heavy constant mu (IGHM) in B-cells [309] is regulated via CSTF2 elevation promoting usage of the weaker proximal internal PAS [29].

CSTF2 up-regulation is associated with several cancers. In an examination of seven tumour types, CSTF2 was found to be up-regulated in five (lung adenocarcinoma, uterine corpus endometrioid carcinoma, bladder urothelial carcinoma, lung adenocarcinoma, and breast invasive carcinoma) and to exhibit a significant correlation between expression and 3′UTR shortening [141]. Additionally CSTF2 is over-expressed in lung cancer, where the abundance correlates functionally with shortening of 3′UTRs and poor prognosis [128,129]. The significance of CSTF2 here is supported by the observation that depletion of CSTF2 suppresses growth, while over-expression promotes growth and invasion [129]. In urothelial carcinoma of the bladder, CSTF2 over-expression results in recruitment to GUAAU motifs at the proximal PAS of the Rac family small GTPase 1 (RAC1) promoting their use and subsequent 3′UTR shortening [89]. Through escape from miRNA-targeted repression, this short isoform exhibits substantially up-regulated RAC1 expression and plays an essential oncogenic role in pathogenesis [89].

Other CstF components have also been shown to be dysregulated in cancer. In B-cell leukemia/lymphoma samples CSTF3 protein is expressed at significantly higher levels in tumours is associated with 3′UTR shortening [87]. In contrast, in triple-negative breast cancer tumours 3′UTR shortening correlating with elevated expression of CSTF3 has been observed [92]. Here increased CSTF3 expression triggers APA shortening of both NRAS proto-oncogene, GTPase (NRAS) and Jun proto-oncogene, AP-1 transcription factor subunit (JUN), oncogenes previously shown to be deregulated in breast cancer [310,311].

Finally, alterations in CstF have been observed in other disorders. In a transverse aortic constriction mouse model of cardiac hypertrophy 3′UTRs were found to be generally shortened and the expression of CstF components CSTF1, CSTF2, and CSTF3 were found to be up regulated [161]. Thus, although the quantitative effects of the CstF complex on APA are not as strong compared to other components of the CPA machinery [13] (Figure 1C), expression changes can be functionally most significant.

#### 2.2.4. Alterations of the CFII Complex in Alternative Polyadenylation and Disease

Although CFII is the least characterised among the CPA complexes, and interacts only weakly and/or transiently with the CPA complex [14], it contributes to PAS recognition via relatively non-specific binding to G-rich far-downstream elements [34]. This binding appears to be via PCF11, as in yeast PCF11 has been mapped downstream from the cleavage site [312]. In contrast, CLP1 is required for the cleavage activity of CFII [34]. Depletion of CLP1 and PCF11 in breast cancer cells demonstrates an overlapping requirement for both proteins in proximal PAS selection [313].

PCF11 is a sub-stoichiometric component of CPA, with levels an order of magnitude lower than other CPA components [314]. It contains evolutionary conserved, tandem canonical PASs within its first intron, which exhibit enriched PCF11 binding, suggesting this low expression could be due to autoregulation by APA and premature termination [314,315]. This suggests even small fluctuations in PCF11 may impact CPA. Indeed, depletion of PCF11 in human cells has been shown to result in significant down-regulation of proximal PAS usage [13,78,79,312,314] (Figure 1C).

In an extensive screen of potential CPA effectors in neuroblastoma cells (Figure 1C) PCF11 emerged as a critical regulator of APA [13]. Here PCF11 depletion resulted in wide ranging transcript lengthening, suggesting a counteraction of APA repression at proximal sites executed by CPSF6. PCF11-mediated APA targets genes with a role in WNT-signalling, influencing cell cycle, proliferation, apoptosis, and neurodifferentiation [13]. Significantly, in neuroblastoma low-level PCF11 expression, significantly fewer adverse outcomes and spontaneous tumour regression are observed, suggesting dysregulated APA can mimic oncogenic mutational events (Figure 3) [13,316]. Accordingly, over-expression of PCF11 directly blocks a physiologically relevant neurodevelopmental program, eventually giving rise to a malignant phenotype. This is supported by observations in large-scale screens where mutations in the PCF11 promoter and 5′UTR, (which possibly can affect expression) have been identified as potential cancer drivers in multiple cancers [317,318].

While the tumour phenotype described above is driven by the pervasive functional impact of PCF11 on APA, there are functionally more complex pathomechanisms illustrating the intricate molecular nature of the CPA machinery. PCF11 also appears to be involved in APA shortening in cancer. Aberrant expression of the ubiquitin ligase adapter MAGE family member A11 (MAGE-A11) in tumours, including prostate [319,320] and breast [321,322] cancers, promotes ubiquitination and proteosomal dependant degradation of PCF11 leading to the loss of NUDT21 from the CPA complex, and ultimately resulting in shortening of transcripts that have enrichment of NUDT21 binding sites upstream of their distal PASs [139].

Finally, outside of cancer, loss of CLP1 results in accumulation of tRNA fragments, which are thought to provoke neurodegenerative disorders [323,324].

#### 2.2.5. Alterations of the CPSF Complex in Alternative Polyadenylation and Disease

CPSF recognises the PAS hexamer [14,19,20] with FIP1L1 binding the U-rich USE [20,25]. Different sub-complexes may be specific for the cleavage or the polyadenylation steps, with CPSF1, CPSF4, WDR33 and FIP1L1 forming a stable complex independently of CPSF2 and CPSF3 [22]. Along with CPSF1 and 4, FIP1L1 levels have a significant impact on APA [13,79]. While distal PASs tend to be stronger [60], for genes with a strong canonical proximal PAS and a short distance to the distal PAS, shortening with FIP1L1 down-regulation has been observed [79,325]. This suggests FIP1L1 levels, along with distance and relative strengths of PASs within a 3′UTR, influence APA. Non-CPA RNA binding factors also regulate PAS selection by CPSF. Cytoplasmic polyadenylation element binding protein 1 (CPEB1), binding to elements upstream of CPSF, may recruit CPSF to weaker DSEs increasing proximal PAS usage, leading to widespread 3′UTR shortening [326]. 3′UTR shortening has been reported to increase stability and promote growth rate of acute myeloid leukemia (AML) cells [86]. In bone marrow mononuclear cells from AML patients, elevated CPSF1 expression was associated with proximal PAS usage in one cellular subtype [86].

Interestingly, although CPSF consists of a large number of factors, each presumably open to mutation, reports of alterations in this complex in disease are lacking. Although reasons for this are currently not clear, potential mechanisms such as functional redundancy within the complex may account for this observation.

#### 2.2.6. Alterations of Poly(A)Polymerases and Poly(A) Binding Proteins in Alternative Polyadenylation and Disease

Polyadenylation of nascent RNA by canonical (PAPα and PAPγ) and non-canonical (terminal uridylyl transferase 1, U6 snRNA-specific (TUT1) or Star-PAP) poly(A) polymerases plays a key role in PAS selection and impacts APA genome-wide [13]. For example, depletion of the canonical PAPs has been shown to result in up-regulation of distal PASs, while for Star-PAP down-regulation of distal PASs with an accompanying up-regulation of intronic PASs is observed [138]. Star-PAP selects mRNA targets for polyadenylation [327] and is required for both the cleavage and polyadenylation steps [328]. Location and surrounding sequence motifs of a PAS also appear to differentiate PAP regulation [138]. In genes with a single PAS, Star-PAP binds upstream of the PAS and recruits CPSF in response to stresses [327,328,329].

Star-PAP has been shown to affect APA in cardiac hypertrophy through an association with the RNA binding protein RNA binding motif protein 10 (RBM10), which stimulates Star-PAP polyadenylation activity [160]. Down-regulation of RBM10 during cardiac hypertrophy and heart failure controls expression and CPA of cardiac mRNAs through the promotion of Star-PAP use of distal PASs and increased poly(A) tail length [159].

Additionally, poly(A) binding proteins are associated with disease. While initially believed to be important for only the polyadenylation step, PABPN1 has been shown to also play a role in the regulation of PAS choice [201,330]. PABPN1 binds proximal PASs as well as affecting APA selection through its role in RNA stability [79,201,330]. PABPN1 knockdown in mouse myoblast cells as well as in human bone osteosarcoma epithelial cells has been found to result in global APA shortening [79,201]. A pan-cancer bioinformatic analysis found PABPN1 expression significantly correlates with distal PAS usage in many types of human cancer, suggesting it may broadly promote usage of distal APA sites in cancer [145]. 3′UTR shortening and lower expression of PABPN1 also correlates with an enhanced proliferative state of lung cancer cells and poor patient prognosis [126].

Oculopharyngeal muscular dystrophy (OPMD) is caused by expansion of a triplet repeat in the PABPN1 gene [331] resulting in nuclear protein aggregations [332]. In a mouse model of OPMD an extensive enhancement of proximal PAS usage was observed [201]. Additionally, depletion of PABPN1 levels in mice results in a consistent decline in distal PAS use and induced muscle wasting [333]. PABPN1 expression levels decline from midlife onwards in skeletal muscle cells and reduced PABPN1 levels correlate with muscle symptoms in OPMD [334]. Altogether, even perturbations of components with seemingly minor effects on APA [13] can have profound functional consequences and result in disease.

#### 2.2.7. Combined Perturbations Affecting Alternative Polyadenylation and Resulting in Disease

In colorectal cancer patients, 3′UTR shortening has been observed to correlate with disease stage [107,110]. An examination of CPA factors revealed NUDT21, CPSF3 and CSTF2 to be over-expressed, while PCF11 and PABPN1 were suppressed in colon cancer tissue [107]. Through combining miRNA profiles with global APA site states in colorectal cancer patients, down-regulation of miR-1-3p was found to regulate APA by up-regulating NUDT21 and CPSF3 [107].

Regulation at the 3′UTR is an important mechanism in heart function and further conditions, including cardiac hypertrophy and heart failure [164]. In dilated cardiomyopathy, even though a significant number of genes exhibited APA, equal proportions showed shifts toward distal and proximal PAS usage. Here, examination of CPA factor expression showed PABPN1 and CPSF4 to be down-regulated, while PCF11 was up-regulated [164]. Although combined alterations of CPA components have not been systematically studied, a recent screening suggests a functional hierarchy of core components of the CPA [13]. This revealed a functional dominance of selected processing factors, for example modulation of PCF11 abundance overrides APA effects executed by other APA regulators.

The previous sections show that, while CPA and APA impact a wide variety of diseases, certain entities, particularly cancers and neurological diseases, appear to be broadly affected. While this may simply reflect biases in investigation, it is tempting to speculate that underlying molecular characteristics may predispose certain tissues to a susceptibility to APA alterations. Given the increasing evidence that CPA and APA impact disease, these findings call attention to the need for research to determine where APA alterations drive disease, and in which cases they simply reflect disease-dependent cellular states.

## 3. APA in Molecular Diagnostics

High-throughput sequencing technologies have significantly promoted the elucidation of disease mechanisms, and have equipped us with novel diagnostic opportunities. As mentioned previously, dynamic changes at the transcriptome 3′ end are prevalent [64,144,154,335] and are commonly associated with differentiation and dedifferentiation processes [336]. Although APA affects more than 70% of all genes, dynamic changes at the transcriptome 3′ end are difficult to detect by standard high-throughput profiling techniques [141,337,338]. APA perturbations are associated with various disorders [78,86,99,162,195,206]. However, they can also possess direct disease-eliciting activities, act as oncogenic drivers, and thereby mimic genetic alterations [13,103]. Importantly, such changes are generally missed in genome profiling endeavours [147]; however, often they also remain undetected by standard RNAseq technologies [339].

APA perturbations, even when resulting in subtle changes of non-coding RNA sequence elements in the 3′UTR, can be functionally most significant [13]. At the same time they also represent unexpectedly potent novel biomarkers [13,141] (Figure 4). For example, in accordance with the functional role of deregulated expression of individual APA-regulators (e.g., high level PCF11 expression supporting proximal polyadenylation of a neurodifferentiation operon; Figure 3) the protein abundance of such APA-regulators can serve as proxy for disease severity and allows stratifying patients (Figure 4, left panel). As one would expect, this prognostic potential can also be found when analysing the resulting downstream consequences i.e., APA-signatures (Figure 4, middle panel). For example, focusing on a select set of genes in which APA is regulated in a PCF11-dependent manner, the relative proportion of long versus short transcript isoforms functions as a proxy for PCF11-dependent APA deregulation. As such, this can be used as a biomarker to predict the patients’ outcome (in this case in neuroblastoma patients). Interestingly, applying receiver operating characteristics (ROC) curve analysis, these APA-signatures appear to perform far better than common clinically used biomarkers (Figure 4, middle panel) and may thus have potential to directly inform clinical decisions [13]. Furthermore, while the mere gene expression change of a given APA-affected gene may not necessarily be predictive, the APA signature of exactly the same genes (abundance long versus short transcript isoform) shows very high predictive potential (Figure 4, right panel). Altogether, these findings have several implications: (1) APA signatures appear to have high, hitherto unused, diagnostic potential, (2) in some instances APA signatures appear to outperform even existing biomarkers, (3) even genes previously identified to be ‘useless’ for disease stratification may have strong diagnostic potential, when APA signatures of the respective genes are considered, (4) the relative proportion of APA isoforms is ‘internally’ controlled as opposed to expression profiling based on arrays or full RNASeq (which can be dramatically confounded by the way of normalization). APA signatures may thus represent relatively robust biomarkers. In light of the challenges to reliably detect transcriptome 3′ end alterations [339], there is a high demand to further develop technologies that allow the adoption of these untapped diagnostic opportunities.

As for alternative splicing, identifying global APA patterns can likely have wide diagnostic implications [340,341,342]. With the evolution of numerous protocols that rely on 3′ end sequencing technologies [13,25,64,99,100,141,190,201,297,335,337,338,339,343,344,345,346,347,348] the determination of APA isoforms is improving. Unlike gene expression profiling based on arrays or full RNASeq, which can be dramatically confounded by the way of normalization (see above), the relative proportion of APA isoforms is normally internally controlled, thus resulting in relatively robust results. While contamination of the analysed specimen by other cell populations is another inevitable and very common confounder in gene expression analysis (i.e., when revealing the signature of a tumour which is infiltrated by immune cells), APA patterns are likely to be more tissue specific [64], and were found to differ according to tissue type, developmental stage, genotype, or cancer subtype [13,87,154,291,292,343,349,350,351]. Cataloguing these (tissue) specific patterns might therefore allow subtraction of “contaminating” APA signatures from APA signatures of specific disordered tissues of interest. Thus, the determination of APA patterns may open up novel diagnostic avenues which up to this point have turned out to represent challenging aspects of “conventional” gene expression profiling. Finally, characterising tissue specific APA signatures per se may be of immediate diagnostic value e.g., for tracing back and identifying the origin of a given disease lesion (for instance in cases of cancer of unknown primary, CUPs).

With the advent of high-throughput analyses, the bioinformatical workload has increased dramatically. In contrast to total RNASeq, the sequencing restricted to the transcriptome 3′ end directly uncovers the variability and perturbation occurring at the mRNA 3′ end. This has several advantages. Firstly, it drastically reduces the bioinformatical workload. Furthermore, these data are typically not confounded by other variables that complicate the bioinformatical processing of the data (such as alternative splicing). Finally, restricting the sequencing to the last (approximately) 30 nucleotides of the transcriptome opens up interesting (and first and foremost cost-effective) opportunities for multiplexing, while still keeping a high coverage for a reliable analysis. In depth APA profile studies have recently revealed “aberrant” APA signatures to be associated with more aggressive tumour phenotypes in cancer patients and thereby provided the proof-of-concept that such a determination can reveal prognostic signatures [13,78,141]. Yet, applying novel bioinformatical analysis (e.g., DaPars), APA patterns can also be extracted from pre-existing transcriptome wide sequencing data [141]. Although this takes advantage of the fact that RNASeq data is already available for numerous tissue specimens, this technique has the limitation that it is primarily suited to detect alternative 3′UTR events, while APA events, which are located within the coding region, or alternatively spliced introns (internal APA) rather remain obscure. Compared to 3′end sequencing technologies this algorithm requires complex bioinformatical calculations, and typically allows a less “intuitive” identification of the mRNA 3′ end [339].

It remains to be observed in which disease conditions and to what extent the analysis of APA signatures could further improve diagnostic strategies and possibly allow detecting biological aberrations with higher sensitivity and specificity. Interestingly, selected APA events can confer strong prognostic power beyond common clinical and molecular variables, suggesting their potential as novel prognostic biomarkers [13,141]. Thus it will be interesting to see how the determination of APA patterns may evolve as a potentially new biomarker in the future. An important key to this development are highly accessible data repositories, which provide insights into the dynamic landscape of APA changes (e.g., TREND-DB; http://shiny.imbei.uni-mainz.de:3838/trend-db/). They provide easy access to APA signatures for non-expert users allowing them to select gene-specific APA signatures to be tested in a targeted approach. This could advance diagnostic strategies for a more thorough understanding of underlying disease mechanisms as well as for a reliable prognostic and possibly therapeutic stratification.

Ultimately, ongoing genome sequencing activities will most likely grant us further insights into genomic variations resulting in gene-specific perturbation of APA isoforms with possible detrimental functional consequences. Unlike global aberration in trans (e.g., as a result of an abundance change of one processing factor or regulatory protein), the cause–consequence relationship in this kind of setting is substantively clearer. Further such changes may be directly accessible for specific, targeted therapeutic approaches.

## 4. Targeting mRNA 3′ end Formation as a Novel Therapy

The significant role that disturbances in CPA play in the diseases presented above reveals the potential of novel therapeutic strategies targeting these mechanisms (Figure 5). How such strategies would be implemented will very much depend on the particular underlying cause in each pathology. For alterations occurring in cis elements, approaches to manipulate CPA are available, although not yet to the clinical level. This is particularly evident in the targeting of specific PASs misused in pathology using antisense oligonucleotides (ASOs).

ASO therapies involve the targeting RNA via complementary base pairing of oligonucleotides. Initial ASOs were capable of recruiting RNaseH resulting in target RNA degradation. However, the development of phosphorodiamidate morpholino oligomers blocked this activity, improved stability, and lowered toxicity [352]. This allowed ASOs to be used to modulate RNA function by blocking access of the cellular machinery to the RNA [353]. ASOs have been used to inhibit translation, modulate splicing, and inhibit miRNA binding [354], and a substantial amount of preclinical data has been produced with many studies reaching clinical trials and even treatment [355].

Strategies using ASOs to block access of the CPA machinery to specific gene PASs to modulate CPA and APA have been developed. As described above, FSHD can result from SNPs creating a canonical PAS resulting in stabilisation of the DUX4 transcript and over-expression. ASOs targeting this novel PAS successfully interfered with CPA resulting in mRNA reduction in differentiated immortalized FSHD myotubes [356,357], as well as in vivo in patient muscle xenografts in immunodeficient mice [357]. A similar approach has been used in prostate cancer, where constitutively active androgen receptor variants lacking the ligand binding domain arise from the use of an intronic PAS [358,359]. ASOs targeting this PAS restored expression of the full-length androgen receptor and inhibited androgen-independent proliferation [359]. When applied to genes exhibiting APA, PAS blocking ASOs may be also used to increase, rather than decrease, expression. For example, the 3′UTR of selectin E (SELE) contains three PASs, and use of a predominant, proximal site yields a shorter transcript missing several destabilizing elements present and results in increased protein levels [360]. Inhibiting use of the distal PAS with ASOs shifted CPA to the proximal sites, resulting in shorter transcripts which exhibit increased mRNA stability ultimately leading to elevated protein expression [361].

CPA is tightly bound to splicing and can be regulated by splicing factors [362,363]. Small nuclear ribonucleoproteins (snRNP) are well characterised for their functions in splicing [364,365], however U1snRNP possesses a splicing-independent inhibitory function on CPA [366] (Figure 5). Through the use of ASOs to block snRNA binding the pre-mRNA, U1snRNP has been shown to play a significant role in suppressing the use of intronic cryptic PASs [367]. This role of U1snRNP has been harnessed to control APA-regulated gene expression (Figure 5). Receptor tyrosine kinases use upstream intronic PASs to generate soluble isoforms lacking the anchoring domains and can act as dominant-negative regulators of signalling pathways [368]. ASOs blocking a 5′ splice site were used to switch isoforms by forcing retention of a PAS-containing intron. This activated the PAS, resulting in expression of the truncated, soluble form of KDR kinase insert domain receptor (VEGFR2) and inhibiting angiogenesis [369].

Additionally it has been demonstrated that through the use of bifunctional U1 adaptors to tether U1snRNP upstream of a PAS, CPA at that PAS can be inhibited [370,371,372]. These adapters consist of a target domain complementary to the target gene and a U1 domain that binds to the U1snRNP. This allows the inhibition of polyA polymerase activity through recruitment of snRNPU1 subunit 70 (SNRNP70) [373]. When targeted to the more canonical PAS this will result in a reduction of gene expression, however this appears to have substantial off target silencing effects [374]. In combination with RNAi this technique has been shown in vivo to result in stronger inhibition than that obtained using either of the techniques alone [370,375].

In addition to ASO targeting of PASs, it has been reported that siRNA targeting can influence CPA. Here an siRNA binding a proximal PAS of the interleukin 4 receptor was found to be active independent of any argonaute RISC catalytic component (Ago) or other RNA-induced silencing complex (RISC)-associated proteins, and appeared to reduce targeted message in an APA-independent manner through deadenylation or inhibition of polyadenylation and subsequent degradation of the immature mRNA [376].

For alterations occurring in cis, the PAS structure of the affected gene is critical. In pathologies resulting from a gain of function a therapeutic approach could be simply to block the novel PAS directly using ASOs or via bifunctional U1 adaptors. If novel PAS occurs within an intron ASO, blocking of splice junctions may be possible. Similar approaches can be taken for genes with multiple PASs, when blocking a proximal PAS may assist in restoring 3′UTR length to genes with mutated distal PAS and vice versa.

However, for conditions where a gene possessing a single PAS is affected, resulting in reduction of expression, options are more limited. Here techniques aimed at correcting the underlaying mutation may be possible. Induced pluripotent stem cells from different cell types from β-thalassemia patients have been used to correct coding region mutations in the HBB using the CRISPR-Cas9 technology [377,378,379]. These techniques would also be applicable for 3′UTR mutations. In an alternative approach, in myotonic dystrophy, transcription activator-like effector nuclease (TALEN)-based insertion has been used to place PASs upstream of repeats in the DM1 protein kinase (DMPK) gene, leading to premature cleavage of transcript before the transcription of the toxic region [380,381]. Editing at the mRNA level is also possible through the use of trans-splicing technology to incorporate sequences into target RNA molecules [382]. This has been successfully used in vivo to correct mutations [383]. Although these technologies have not been used to correct CPA abnormalities, it is conceivable that similar approaches could be used to replace defective PASs.

While clear approaches are available for the correction of specific gene defects resulting in pathological CPA, the treatment of global APA alteration provides more challenges. However, avenues exist where the CPA machinery itself may be a potential therapeutic target. The availability of CPA factors has a significant effect on APA [13]. Systematic depletion of individual CPA components has been shown to influence APA [25,79]. As described above, CPA factors are associated with the progression of multiple human cancers, potentially functioning as tumour suppressors [384,385] or oncogenic factors [129,386,387,388], suggesting they may serve as potential therapeutic targets. An example of potential therapy is shown by depletion of CPSF1. This factor suppresses ovarian cancer cell growth and proliferation in vitro [151], reduces expression of pathogenic APA-induced androgen receptor variant, and up-regulates expression of the full-length androgen receptor in prostate cancer cells [359]. Other factors potentially targetable in a similar manner include CFI components PCF11 and FIP1L1, which upon depletion enhance distal PAS usage [13,78]. A final example is found in idiopathic pulmonary fibrosis development, where transforming growth factor beta 1 (TGFB1) mediates the down-regulation of NUDT21 via induction of miR-203 [389] resulting in APA shortening [195]. This suggests targeting of either the TGFB1 or NUDT21 may rectify APA and provide a therapy.

Other cellular processes controlling gene expression influencing APA may serve as therapeutic targets. One such potential target stems from the observation that processivity of RNA Pol II can influence APA, and mutations in RNA Pol II directly slowing down RNA Pol II elongation favour proximal PAS use [70]. Therefore, manipulation of RNA Pol II may provide a means of influencing global APA direction. This is particularly relevant as several anticancer drugs, such as doxorubicin or camptothecin, impact RNA Pol II processivity [390]. Additionally, extensive post-translational phosphorylation of RNA Pol II influences co-transcriptional events including splicing, transcription termination, and 3′ end processing [391], providing potential targets for APA manipulation. Another example connecting post-translational modifications with the regulation of 3′ end processing as shown for PAP may lead to new avenues in targeting signalling components for regulation of the transcriptome 3′ end diversity [392].

Just as a diverse range of compounds targeting the spliceosome have been developed [393,394,395,396,397], chemical modulators of CPA and APA are also available. Cordycepin, a well-known chemotherapeutic drug, inhibits CPA [398,399] by incorporating into the poly(A) tail trapping the CPSF complex [14,400,401]. Cordycepin has also been studied in numerous cell lines including oral cancer [402], cervical and breast cancer [403], lymphomas and leukemias [404], and multiple myeloma [405]. Enhanced proliferation and reduced apoptosis susceptibility are characteristics of endometriosis. Cordycepin suppresses proliferation and activates apoptosis in human epithelial endometriotic cells in vitro [406]. Osteoarthritis patients exhibit elevated CPSF4 expression, resulting in high levels of inflammatory genes [407]. Cordycepin has also been shown to reduce inflammatory gene induction in cell culture [408], and symptoms in rodent models of osteoarthritis [407].

Additionally, as epigenetic modifications such as genomic imprinting [54], DNA methylation [54,409], and histone modification [52] can control APA, it is tempting to speculate that the manipulation of these pathways may eventually be translated into the clinical context.

Therapeutic intervention can also be achieved on the front of RBPs, which in the future may form the basis of therapies to influence CPA [410]. For example: CPEB1 mediates shortening in coordination with mRNA translation [411]; the FUS RNA binding protein frequently binds 3′UTRs enhancing CPA at upstream PASs and down-regulating downstream PASs resulting in short transcripts [210]; in Drosophila embryonic lethal abnormal vision (ELAV) directly binds proximal PASs facilitating shorter isoform expression [412]; and finally depletion of MBNL results in altered APA [200].

Finally well-established drugs may have unexpected effects on CPA. As described above, APA of SLC6A4 plays a role in several neurological conditions and interestingly treatment of mice with the SLC6A4-selective antidepressant/anxiolytic drug fluoxetine, which has been shown to reduce anxiety disorders, increases expression of the distal polyadenylation form of SLC6A4 [216,413]. With the development of novel CPA inhibitors [414] this therapeutic avenue holds great promise.

## 5. Conclusions and Perspectives

Post-transcriptional gene regulation through mRNA 3′UTR sequences has emerged as a critical process controlling important cellular functions by directing transcript stability, nuclear export, sub-cellular localisation and translation efficiency. In recent years the advent of high-throughput sequencing has revealed the enormous contribution CPA and APA plays in sculpting the 3′UTR. Far from acting as a constant mechanism simply terminating mRNA transcription, CPA has emerged as a complex process, tightly integrated with other transcriptional processes regulating the diverse molecular aspects of mRNA metabolism.

APA has been found to represent an important layer of post-transcriptional gene regulation, playing a central role in the establishment of regulated expression networks fundamental to a multitude of biological roles. Modulation of APA in various normal physiological circumstances can influence RNA fate and regulate protein output both quantitatively and qualitatively, directing important cellular programs. Regulation of CPA relies on the precise integration of transcription, with trans-acting CPA factors interacting with specific cis-elements in the pre-mRNA. For individual genes where mutations in these cis elements disturbs CPA, the relationship to pathology is clear. However, although altered APA signatures have been associated with a variety of disorders, the contribution of CPA factors on global APA in human pathologies is still unclear. First evidence is accumulating [13] that APA can also represent a driver in human pathology.

Given this increasing recognition of the prevalence of CPA and APA, further studies defining key components directing these processes are required to decipher their functional contribution and significance in normal cellular processes as well as human disease. Additionally, an understanding of the cellular pathways that regulate CPA and APA and how these molecular mechanisms connect are needed to understand the development of certain pathologies. Finally, in order to determine whether APA can serve as a meaningful therapeutic target, biological model systems as well as comprehensive analysis tools for APA are required.

This review highlights the relevance of CPA and APA mechanisms in the correct expression of genes involved in disease development and cell homeostasis, and how this information may impact diagnosis and therapy. It will be interesting to see how these findings translate into the manipulation of APA in clinical settings and result in the identification of disease biomarkers that may be used as diagnostic and therapeutic tools in the future.

## Figures and Tables

**Figure 1 biomolecules-10-00915-f001:**
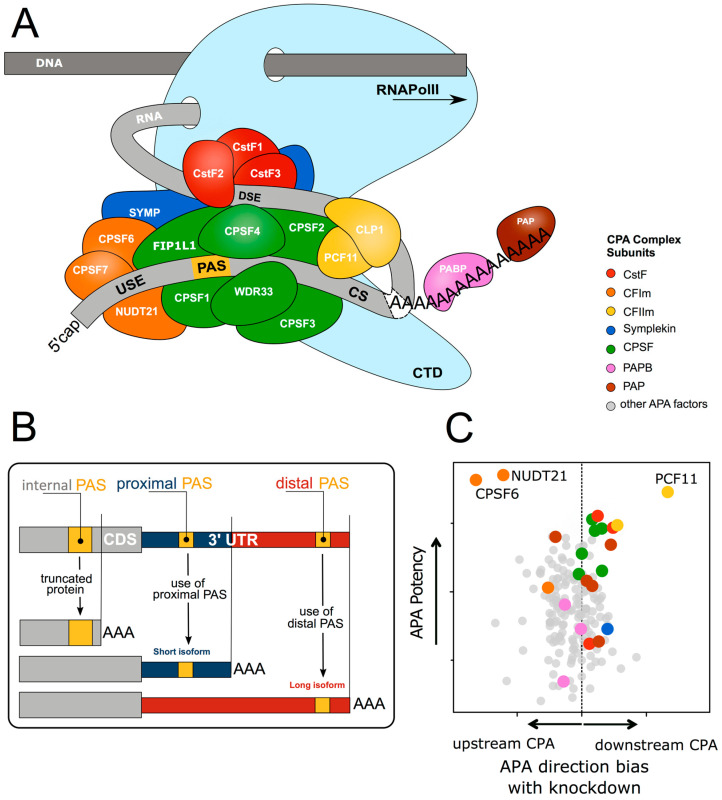
The core 3′ end RNA processing machinery and impact on alternative polyadenylation. (**A**) The core 3′ end processing machinery consists of complexes composed of multiple trans acting proteins interacting with RNA via multiple cis-elements (USE = upstream sequence element; PAS = poly(A) signal; CS = cleavage site; DSE = downstream sequence element; CTD = C-terminal domain). Upon co-transcriptional assembly of these complexes, RNA cleavage and polyadenylation occurs to form the 3′ end of the nascent RNA molecule. (**B**) More than 70% of all genes harbour more than one polyadenylation signal (PAS). This gives rise to transcript isoforms differing at the mRNA 3′ end. While alternative polyadenylation (APA) in 3′UTR changes the properties of the mRNA (stability, localisation, translation), internal PAS usage (in introns or the coding sequence (CDS)) changes the C-termini of the encoded protein, resulting in different functional or regulatory properties. (**C**) Abundance of individual 3′ end processing components results in variable impacts on level and direction of APA (compare [13]). Shown are effects on global PAS choice after siRNA depletion of more than 170 potential APA regulators, including the highlighted core CPA components (color code corresponds to complexes depicted in Figure 1A; Y-axis shows effect size, X-axis shows directionality of global APA bias upon depletion of individual factors (further information see Supplementary Figure 2 [13])). While depletion of most factors does not show any directional APA bias, depletion of the CFIm (NUDT21 and CPSF6) and CFIIm (PCF11) complex components pervasively regulate APA in a unidirectional manner (i.e., resulting in primarily longer or shorter transcript isoforms after depletion of PCF11 or NUDT21, and CPSF6 respectively) Further data can be found in TREND-DB (http://shiny.imbei.uni-mainz.de:3838/trend-db/).

**Figure 2 biomolecules-10-00915-f002:**
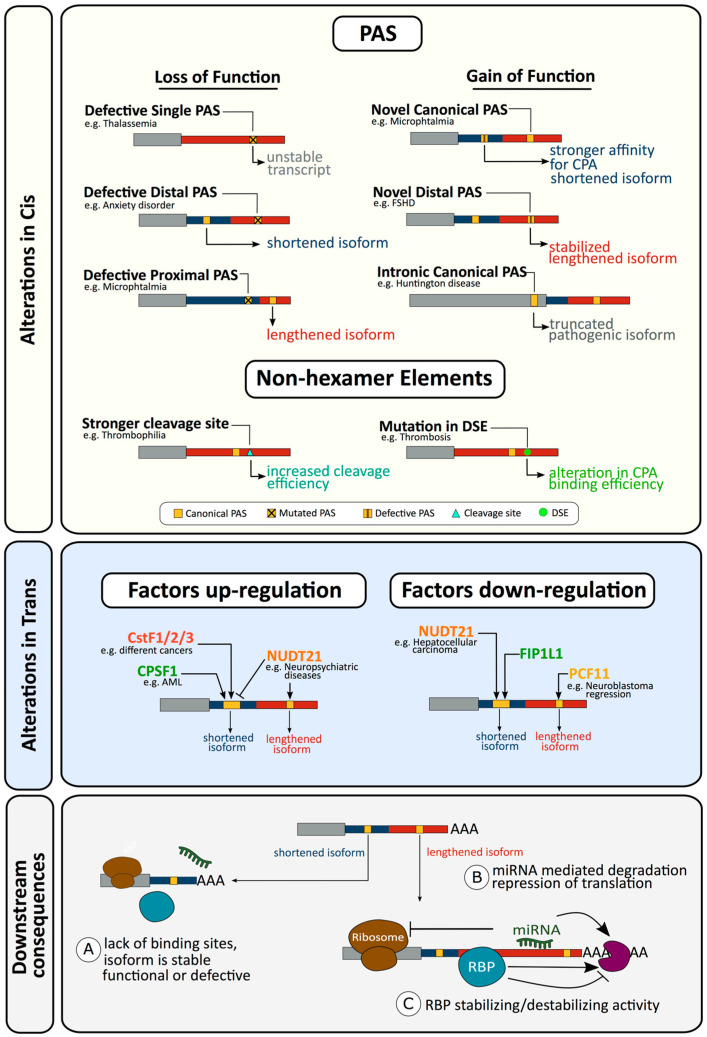
Examples of CPA and APA alterations resulting in disease. Alterations can occur in cis (top panel) or in trans (middle panel) resulting in differing 3′UTR isoforms which in turn offer different platforms for regulation of expression (bottom panel, further details see text and Table 1).

**Figure 3 biomolecules-10-00915-f003:**
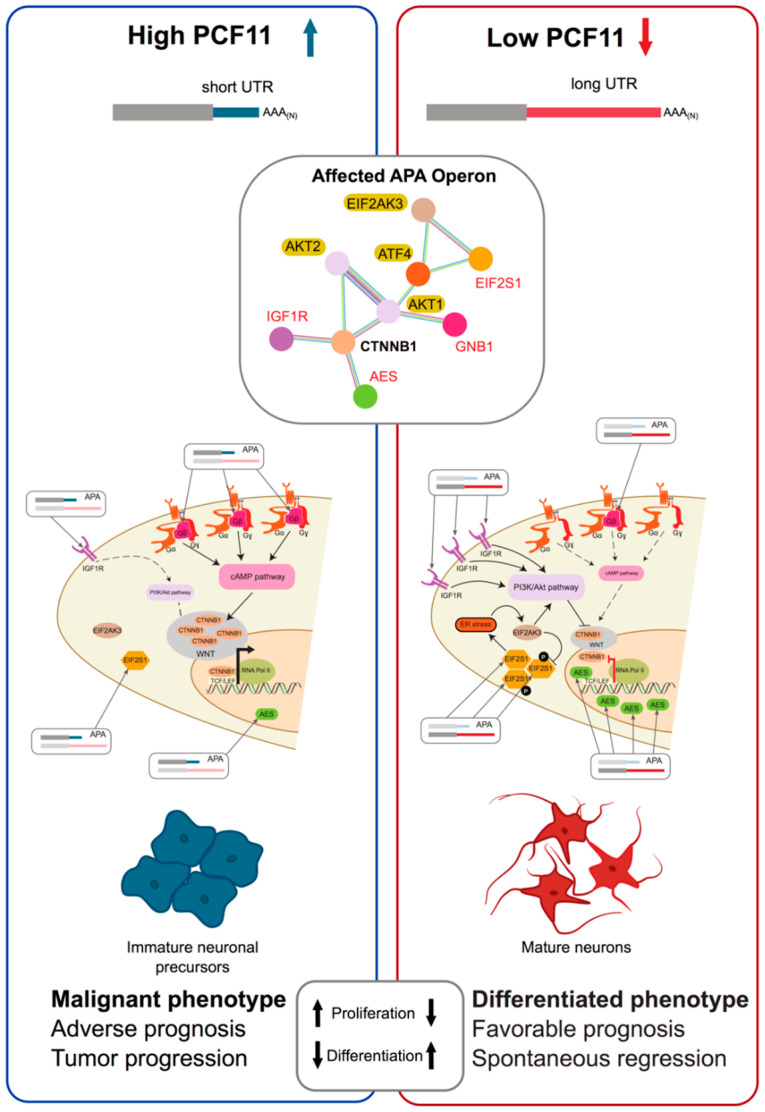
PCF11-directed APA drives neuroblastoma. An example of APA perturbation eliciting deleterious cellular programs. PCF11 is among the top drivers of APA [13], compare TREND-DB (http://shiny.imbei.uni-mainz.de:3838/trend-db/). Postnatal down-regulation results in general 3′UTR lengthening required to induce neurodifferentiation. PCF11-directed APA targets insulin like growth factor 1 receptor (IGF1R), eukaryotic translation initiation factor 2 subunit alpha (EIF2S1), TLE family member 5, transcriptional modulator (AES, TLE5) and G protein subunit beta 1 (GNB1), which constitute a highly enriched APA-operon (Red; String-DB), converging on WNT signalling via beta-catenin (CTNNB1; bold). This alters downstream IGF1R, phosphatidylinositol-4,5-bisphosphate 3-kinase (PI3K)/AKT serine/threonine kinase and endoplasmic reticulum (ER) stress response signalling pathways (highlighted), ultimately modulating cell cycle progression, proliferation, apoptosis and neuronal differentiation. Sustained high level PCF11 expression arrests neuronal precursors in an immature state, giving rise to neuroblastomas. Low-level PCF11 in neuroblastoma associates with favourable outcome and spontaneous tumour regression, while high-level PCF1 associates with adverse outcome and tumour progression (other genes indicated eukaryotic translation initiation factor 2 alpha kinase 3 (EIF2AK3), AKT serine/threonine kinase 2 and 2 (AKT2 and 2), activating transcription factor 4 (ATF4), eukaryotic translation initiation factor 2 subunit alpha (EIF2S1), catenin beta 1 (CTNNB1); figure modified after A. Ogorodnikov, Dissertation, Johannes Gutenberg University Mainz, 2017 “Massive RNAi screening dentifies key drivers of transcriptome 3′end diversity with a direct role in neuroblastoma tumor regression”).

**Figure 4 biomolecules-10-00915-f004:**
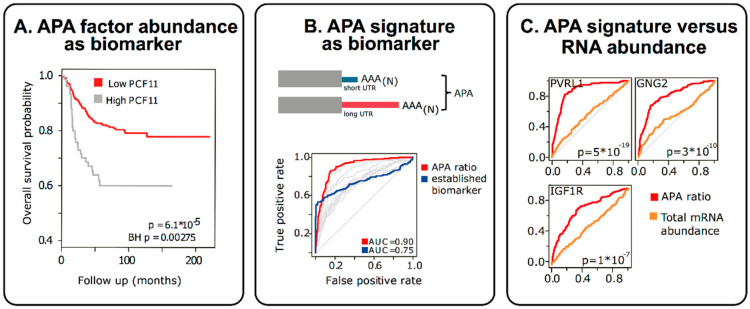
APA signatures are potent biomarkers (example shown for neuroblastomas). (**A**) High level PCF11 associates with lower survival probability. (**B**) PCF11-directed APA changes (APA ratio) represent potent prognostic biomakers (a current commonly used biomarker is shown in blue). In contrast, mere mRNA abundance changes of APA-affected targets (e.g., PVRL1, GNP2 or IGF1R) do not possess diagnostic potential (**C**). Thus, even genes with previously identified insignificant RNA abundance changes may possess strong diagnostic potential when qualitative aspects (APA ratio reflecting different transcript isoforms) are considered.

**Figure 5 biomolecules-10-00915-f005:**
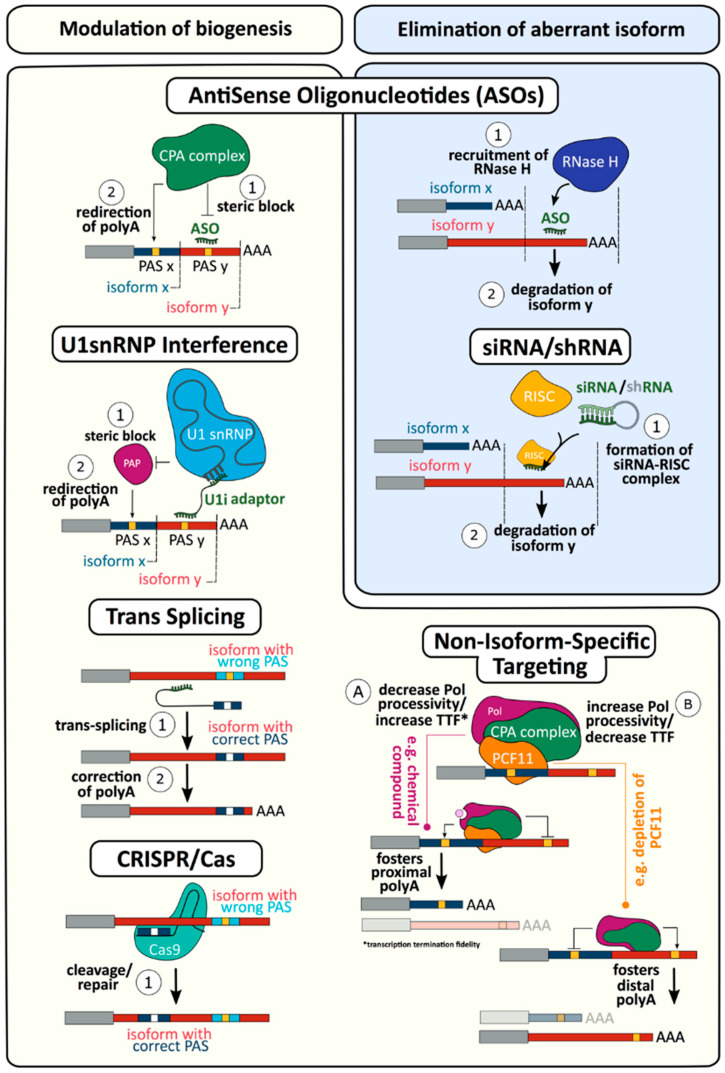
CPA/APA and therapy. Different approaches and mechanisms to correct deregulated APA through modulation of biogenesis and/or elimination of faulty isoforms. *Antisense Oligonucleotides (ASOs)*: A designed sequence of oligonucleotides anneals a target region, leading alternatively to the steric block of the CPA complex or recruitment of RNAseH and subsequent degradation. The choice of one mechanism over the other depends on the generation of nucleotides used. *U1snRNP interference*: A bifunctional U1 adaptor is designed to anneal simultaneously to the 5′ end of U1snRNP and to the target sequence. The steric block induced by the complex redirects PAP to a different PAS. *Trans Splicing*: The therapeutic agent comprises a binding domain complementary to the target pre-mRNA, an unpaired oligonucleotide spacer and a coding domain containing the sequences to be trans-spliced into the target. This structure enhances specific trans-splicing of the target pre-mRNA, which competes with the cellular cis-splicing (which would bring the “pathological” mRNA) by creating a chimeric RNA molecule. The exogenous sequence carried by the structure is subsequently incorporated in the mature transcript, downstream enabling the biosynthesis of a significant amount of the corrected protein. *CRISPR/Cas9*: By means of Cas9 enzyme, the isoform containing the wrong PAS can be edited, inserting a new sequence containing the correct PAS. *siRNA/shRNA*: The administered siRNA/shRNA, after being processed in the nucleus, is loaded on RISC. The nucleotide strand directs RISC to the target mRNA leading to degradation or silencing of the complex. *Non-isoform-specific targeting*: The choice of one isoform over the other can be fostered, interfering with the CPA machinery. (A) Administration of chemical compounds targeting PolII can decrease processivity of the enzyme and increase transcription termination fidelity. Contrary to that, (B) depletion of specific components of the machinery (e.g., PCF11) leads to an opposite result. Interference based on “component-targeting” rather than “isoform-targeting” reflects into widespread changes eliciting possible side effects.

**Table 1 biomolecules-10-00915-t001:** CPA/APA and human disease.

Disease	Type	Effector Factors	Cis/Trans	Gain/Loss	References
**Cancer**					
Acute myeloid leukaemia	APA	CPSF	Trans	Unknown	[86]
B-cell leukemia/lymphoma	APA-S	CFII, CPSF, CstF	Trans	Unknown	[87]
Bladder	APA-S	CFI, CstF	Trans	Unknown	[88,89]
Breast	APA-I	Estrogen	Trans	Unknown	[90,91]
Breast	APA-S	CstF, Estrogen	Trans	Unknown	[92,93,94,95,96,97,98,99,100,101,102]
Chronic lymphocytic leukaemia	APA-I	Unknown	Trans	Unknown	[103]
Colon	APA-S	Unknown	Trans	Unknown	[104,105]
Colorectal	APA-S	CPSF, CFI, hnRNPC	Trans	Unknown	[106,107,108,109,110,111]
Esophageal squamous cell carcinoma	CPA	PAS	Cis	Loss	[112]
Fanconi Anemia	APA-I	PAS-methylation, SF3A1	Trans	Gain	[113,114]
Gastric	APA-S	Unknown	Trans	Unknown	[115]
Glioblastoma	APA-L	Unknown	Trans	Unknown	[116]
Glioblastoma	APA-S	CFI, PTBP1	Trans	Loss	[78,117,118,119,120]
Hepatocellular carcinoma	APA-S	CFI	Trans	Unknown	[121,122,123]
Lung	APA-S	CFI, CstF	Trans	Unknown	[102,124,125,126,127,128,129,130,131]
Lynch syndrome	CPA	PAS	Cis	Loss	[132]
Mantle cell lymphoma	APA-S	PAS	Cis	Gain	[133,134]
Melanoma	CPA	Unknown	Trans	Unknown	[135]
Multiple	APA	UTR3 SNP	Cis	SNP	[136]
Multiple	APA-I	PAS-Pbx1 binding	Cis	Gain	[137]
Multiple	APA-S	CFI, CstF, PAPs, PABPN1	Trans	Unknown	[138,139,140,141,142,143,144,145]
Multiple	CPA	PAS	Cis	Loss	[146,147]
Nasopharyngeal carcinoma	APA	Unknown	Trans	Unknown	[148]
Neuroblastoma	APA-L	CFII	Trans	Loss	[13]
Ovarian	APA-S	Unknown	Trans	Unknown	[149,150,151,152]
Pancreatic ductal adenocarcinoma	APA-S	Unknown	Trans	Unknown	[153]
Proliferative conditions	APA-S	Unknown	Trans	Unknown	[154,155]
Prostate	CPA	Other	Trans	Loss	[156]
Small intestinal neuroendocrine	APA-S	Unknown	Trans	Unknown	[157]
**Cardiovascular**					
Atherosclerosis	CPA	Unknown	Trans	Unknown	[158]
Cardiac hypertrophy	APA	CstF, PAP	Trans	Unknown	[159,160,161,162]
Cardiac hypertrophy	CPA	PABPC1	Trans	Unknown	[163]
Cardiomyopathy, Dilated	APA	CFII, CPSF, PABPN1	Trans	Unknown	[164]
Hypertension	APA-L	UTR3 SNP	Cis	Unknown	[165,166]
Hypertension	APA-S	DSE	Cis	Gain	[167]
Ischemia/reperfusion injury	APA-S	Unknown	Trans	Unknown	[168]
Thrombosis (Deep vein)	APA-I	DSE	Cis	Gain	[169,170]
Thrombosis (Venous)	CPA	CS, DSE	Cis	Gain	[171,172,173,174]
**Congenital Abnormalities**					
Microphthalmia	APA-L	PAS	Cis	Loss	[175]
Mullerian aplasia	CPA	Unknown	Trans	Unknown	[176]
Bone fragility (Osteogenesis imperfecta)	CPA	PAS	Cis	Loss	[177]
Rickets	APA	Unknown	Trans	Unknown	[178]
**Endocrine**					
Diabetes, neonatal	CPA	PAS	Cis	Loss	[179]
Diabetes, type 2	APA	Unknown	Trans	Unknown	[180]
**Hematological**					
Glanzmann Thrombasthenia	CPA	PAS	Cis	Loss	[181]
α-Thalassemia	CPA	PAS	Cis	Loss	[182]
β-Thalassemia	CPA	PAS	Cis	Loss	[183,184]
**Immunological**					
IPEX syndrome	CPA	PAS	Cis	Loss	[185,186]
Nasal polyps	APA	Unknown	Trans	Unknown	[187,188]
Severe combined immunodeficiency	CPA	PAS	Cis	Loss	[189]
Systemic lupus erythematosus	APA-L	PAS	Cis	Gain, Loss	[190,191,192,193]
Wiskott-Aldrich syndrome	CPA	PAS	Cis	Loss	[194]
**Lung Disease**					
Pulmonary fibrosis	APA-S	CFI	Trans	Loss	[195]
**Musculoskeletal**					
Muscle fibrosis	APA-I	Unknown	Trans	Unknown	[196]
Muscular Dystrophy, Facioscapulohumeral	CPA	PAS	Cis	Gain	[197]
Muscular dystrophy, Oculopharyngeal	APA-S	PABP	Trans	Loss	[198,199]
Myotonic dystrophy	APA-S	CFI	Trans	Loss	[200]
Oculopharyngeal muscular dystrophy	APA	PABPN1	Trans	Loss	[201]
**Neurological**					
Alzheimer Disease	APA	Unknown	Trans	Unknown	[202,203,204,205,206]
Alzheimer Disease	CPA	U1 snRNP	Trans	Unknown	[207]
Amyotrophic lateral sclerosis	APA	FUS, TARDBP	Trans	Loss	[206,208,209,210,211]
Amyotrophic lateral sclerosis	CPA	PAS	Cis	Gain	[212,213]
Anxiety Disorders	APA-S	PAS	Cis	SNP	[214,215,216]
Fabry disease	CPA	PAS	Cis	Loss	[217]
Fragile X syndrome	APA-S	UTR5 repeats	Cis	Unknown	[218]
Friedreich’s Ataxia	APA	CPSF	Trans	Loss	[219]
Huntington’s disease	CPA	PAS	Cis	Gain	[220]
Metachromatic leukodystrophy pseudodeficiency	APA-L	PAS	Cis	Loss	[221,222]
Neuropsychiatric disease	APA-L	CFI	Trans	Gain	[223]
Parkinson disease	APA	UTR3 SNP	Cis	Unknown	[224]
Parkinson disease	APA-L	PAS dopamine	Trans	Unknown	[206,224]
Suicidal behavior	APA-S	PAS	Cis	SNP	[225]
**Other**					
Stress	APA	Unknown	Trans	Unknown	[226]
Zellweger syndrome	APA-S	PAS	Cis	Loss	[227]

APA—alternative polyadenylation, APA-S—APA shortening, APA-L—APA lengthening, APA-I—APA internal, CPA—cleavage and polyadenylation, PAS—polyadenylation signal, hnRNPC—heterogeneous nuclear ribonucleoprotein C, SF3A1—splicing factor 3a subunit 1, PTBP1—polypyrimidine tract binding protein 1, Pbx1—PBX homeobox 1, U1 snRNP—small nuclear ribonucleoprotein U1, TARDBP—TAR DNA binding protein.

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
