# Peer review of "Emerging Roles of RNA 3′-end Cleavage and Polyadenylation in Pathogenesis, Diagnosis and Therapy of Human Disorders"

_biomolecules, 2020, doi:10.3390/biom10060915_

Round 1
Reviewer 1 Report
The review is devoted to the very interesting process of cleavage/polyadenylation events in mRNA maturation upon transcription. The authors concentrated on appearance of alternative sites and the influence of the mutations in regulatory elements on pathological processes and development of the disease. The also examined diagnostic potential and possible approaches for targeting 3'-UTR of mRNA as a novel therapy.
The review contains huge amount of data including very recent results from the papers of 2020.
The review will be interesting for broad scientific community.
The only comment is that the figures should be more detailed and should be more figures to make the text more easy to read.
Author Response
Response to reviewer reports
We thank both reviewers for their very supportive and positive critique. We introduced the following changes (detailed below) and hope that the manuscript is now ready for publication.
Reviewer 1
The only comment is that the figures should be more detailed and should be more figures to make the text more easy to read.
Reply: To assist in the readability of the large disease section a new figure (Figure 2) has been introduced summarising the mechanisms by with CPA and APA can result in altered gene expression
Reviewer 2
Figure 1C. The data is from ref. 13 but it would babe nice if the analysis shown in this panel is explained a bit more. It does not directly correspond to any figure panel in ref. 13.
Expanded legend for Figure 1C to include:
Abundance of individual 3' end processing components results in variable impacts on level and direction of APA (compare [13]). Shown are effects on global PAS choice after siRNA depletion of more than 170 potential APA regulators, including the highlighted core CPA components (color code corresponds to complexes depicted in Figure 1a; Y-axis shows effect size, X-axis shows directionality of global APA bias upon depletion of individual factors (further information see supplementary Fig. 2d [13])). While depletion of most factors does not show any directional APA bias, depletion of the CFIm (NUDT21 and CPSF6) and CFIIm (PCF11) complex components pervasively regulate APA in a unidirectional manner (i.e. resulting in primarily longer or shorter transcript isoforms after depletion of PCF11 or NUDT21, and CPSF6 respectively).
Many gene abbreviations are used throughout the manuscript but the full gene names are not stated. I understand that this would add some extra to the character count but I think it would be worth it, given the manuscript has a 'definitive resource' character.
Reply: Full gene names have been included throughout the manuscript. See Word Track Changes
line 71: This is the only time AAUAAA and variants are spelled out with a 'U' instead of a 'T'. Could this be harmonised?
Changed AAUAAA in line 71 to AATAAA
line 91: 'and' instead of 'an'.
revised as requested
Table 1. some of the column header have spilled over into a second line.
revised as requested
line 197: 'alpha' instead of 'a'.
revised as requested
line 571: 'with' instead of 'to'.
revised as requested
line 606: The PAT-seq method could be cited here, doi:10.1261/rna.048355.114
citation included (reference #348)
Reviewer 2 Report
The manuscript by Nourse et al. is a very carefully complied and comprehensive review of CPA/APA and its disease relevance. With its clear structure, useful figures and tables and a long list of cited references, it will be a very useful resource for anyone working in this area, whether they are new to the field and perhaps coming in 'sideways' from a specific medical context, or whether they have a longstanding association with CPA/APA as part of gene expression. I have only very minor suggestions for improvement, as listed below.
Figure 1C. The data is from ref. 13 but it would babe nice if the analysis shown in this panel is explained a bit more. It does not directly correspond to any figure panel in ref. 13.
Many gene abbreviations are used throughout the manuscript but the full gene names are not stated. I understand that this would add some extra to the character count but I think it would be worth it, given the manuscript has a 'definitive resource' character.
line 71: This is the only time AAUAAA and variants are spelled out with a 'U' instead of a 'T'. Could this bee harmonised?
line 91: 'and' instead of 'an'.
Table 1. some of the column header have spilled over into a second line.
line 197: 'alpha' instead of 'a'.
line 571: 'with' instead of 'to'.
line 606: The PAT-seq method could be cited here, doi:10.1261/rna.048355.114
Author Response

(The authors gave the same response as above.)
